# The Novel Mnk1/2 Degrader and Apoptosis Inducer VNLG-152 Potently Inhibits TNBC Tumor Growth and Metastasis

**DOI:** 10.3390/cancers11030299

**Published:** 2019-03-03

**Authors:** Senthilmurugan Ramalingam, Vidya P. Ramamurthy, Lalji K. Gediya, Francis N. Murigi, Puranik Purushottamachar, Weiliang Huang, Eun Yong Choi, Yuji Zhang, Tadas S Vasaitis, Maureen A. Kane, Rena G. Lapidus, Vincent C. O. Njar

**Affiliations:** 1Department of Pharmacology, University of Maryland School of Medicine, 685 West Baltimore Street, Baltimore, MD 21201, USA; SRamalingam@som.umaryland.edu (S.R.); rv.darsini@gmail.com (V.P.R.); lalji.gediya@gmail.com (L.K.G.); njamurigi@gmail.com (F.N.M.); PPuranik@som.umaryland.edu (P.P.); 2Center for Biomolecular Therapeutics, University of Maryland School of Medicine, 685 West Baltimore Street, Baltimore, MD 21201, USA; 3Department of Pharmaceutical Sciences, University of Maryland School of Pharmacy, Baltimore, MD 21201-1559, USA; whuang@rx.umaryland.edu (W.H.); mkane@rx.umaryland.edu (M.A.K.); 4Marlene and Stewart Greenebaum Comprehensive Cancer Center, University of Maryland School of Medicine, 685 West Baltimore Street, Baltimore, MD 21201, USA; echoi@som.umaryland.edu (E.Y.C.); rlapidus@som.umaryland.edu (R.G.L.); 5Division of Biostatistics and Bioinformatics, University of Maryland Marlene and Stewart Greenebaum Comprehensive Cancer Center, Baltimore, MD 21201-1559, USA; Yuzhang@som.umaryland.edu; 6Department of Epidemiology and Public Health, University of Maryland School of Medicine, Baltimore, MD 21201, USA; 7Department of Pharmaceutical Sciences, School of Pharmacy and Health Professions, University of Maryland Eastern Shore, 207 Somerset Hall, Princess Anne, MD 21853, USA; tsvasaitis@umes.edu

**Keywords:** triple negative breast cancer, metastasis, Mnk1/2 degraders, VNLG-152R, apoptosis, Mnk-eIF4E/mTORC1 signaling pathways

## Abstract

Currently, there are no effective therapies for patients with triple-negative breast cancer (TNBC), an aggressive and highly metastatic disease. Activation of eukaryotic initiation factor 4E (eIF4E) by mitogen-activated protein kinase (MAPK)-interacting kinases 1 and 2 (Mnk1/2) play a critical role in the development, progression and metastasis of TNBC. Herein, we undertook a comprehensive study to evaluate the activity of a first-in-class Mnk1/2 protein degraders, racemic VNLG-152R and its two enantiomers (VNLG-152E1 and VNLG-152E2) in in vitro and in vivo models of TNBC. These studies enabled us to identify racemic VNLG-152R as the most efficacious Mnk1/2 degrader, superior to its pure enantiomers. By targeting Mnk1/2 protein degradation (activity), VNLG-152R potently inhibited both Mnk-eIF4E and mTORC1 signaling pathways and strongly regulated downstream factors involved in cell cycle regulation, apoptosis, pro-inflammatory cytokines/chemokines secretion, epithelial-mesenchymal transition (EMT) and metastasis. Most importantly, orally bioavailable VNLG-152R exhibited remarkable antitumor (91 to 100% growth inhibition) and antimetastatic (~80% inhibition) activities against cell line and patient-derived TNBC xenograft models, with no apparent host toxicity. Collectively, these studies demonstrate that targeting Mnk-eIF4E/mTORC1 signaling with a potent Mnk1/2 degrader, VNLG-152R, is a novel therapeutic strategy that can be developed as monotherapy for the effective treatment of patients with primary/metastatic TNBC.

## 1. Introduction

Breast cancer (BC) is the most frequently diagnosed cancer among women in the United States and is the second leading cause of cancer-related death in women [1]. Although early diagnosis and new treatment options are effective for the management of primary breast cancer, treatment of advanced breast cancer, specifically metastatic disease is still a challenge [2]. Triple-negative breast cancers (TNBCs) lack the expression of estrogen receptor (ERα), progesterone receptor (PgR) and human epidermal growth factor receptor 2 (HER2) and are the major cause of BC mortality due to aggressive invasive and metastatic potential; lack of suitable molecular treatment targets; and resistance to conventional chemotherapeutic agents [3,4]. Hence, there is an urgent need to identify and develop new therapeutic drugs that are effective against TNBCs and metastatic breast cancers (MBCs), especially those that can offer higher survival rates, fewer side effects, and a better quality of life for patients than the currently available therapies [5].

Eukaryotic initiation factor 4E (eIF4E), a component of “eIF4F cap binding complex” has crucial roles in mRNA discrimination and in driving the development and progression of various cancers including that of the breast [6,7]. eIF4G-associated eIF4E binds to m7G cap at the 5′-end of eukaryotic mRNAs, initiating translation of “weak mRNAs” encoding malignancy-related proteins [8,9]. Significantly, eIF4E(S209) phosphorylation by mitogen-activated protein kinase (MAPK)-interacting kinase 1 and 2 (Mnk1/2) is important for tumorigenicity, but not for normal mammalian growth [6,10,11]. eIF4E knockdown is shown to decrease BC cell proliferation in rapamycin-sensitive and rapamycin-insensitive BC cell lines [12]. eIF4E depletion also enhanced the anti-proliferative and pro-apoptotic effects of chemotherapeutic drugs in BC cells [13]. Wheater et al. [14] demonstrated that treatment of human breast cancer cells with Mnk1 inhibitors reduced colony formation, proliferation, and survival. The impact of protein translation has also been clearly demonstrated in the clinical setting. For example, a recent prospective study found that >60% of TNBC patients have tumors with high levels of eIF4E, with the conclusion that TNBC patients with high eIF4E overexpression are more likely to recur and die from cancer recurrence and that high eIF4E seems to be a significant prognostic marker in TNBC patients [15]. eIF4E is reported to be essential for BC progression, angiogenesis [16] and metastasis [17]. eIF4E elevation of 7-fold or more is a strong prognostic indicator of BC relapse and death in both retrospective and prospective clinical studies [15,18,19,20]. In addition, recent reports highlight the impact of inhibiting Mnk-eIF4E signaling in experiment and clinical settings of TNBC [21,22,23].

There has been increasing interest in the development of agents that inhibit oncogenic eIF4F translation complex as strategy to obtain effective anticancer therapeutics against a variety of solid tumors and hematologic cancers [24,25,26,27,28,29]. Amongst the various strategies of disrupting eIF4F complex (and as a consequent, eIF4E), inhibition of Mnk1/2 to prevent phosphorylation of eIF4E is generating keen interest, evidenced by the increasing number of research publications and patents/patent applications (reviewed in ref. [30]). The novel and highly significant recent serendipitous finding that Mnk1 and Mnk2 regulate mammalian target of rapamycin complex 1 (mTORC1) signaling and associates with mTORC1 directly [31,32] suggest that Mnk1/2 inhibitors/degraders can also inhibit mTORC1/4E-BP1/p70SK6 signaling. Indeed, our previous studies [33] and other independent studies [34,35,36,37] strongly support the connection between Mnk1/2 and mTORC1 and molecular cross-talk between both pathways. Thus, pharmacological inhibition of Mnk1/2, and, therefore, inhibition of both Mnk-eIF4E and mTORC1 pathways, might serve as a potential effective therapeutic approach for treating patients with advanced breast cancer including TNBCs and MBCs and possibly other malignancies with dysregulated Mnk-eIF4E/m-TORC1 signaling.

Our group recently discovered a class of novel retinamides (NRs; semi-synthetic natural products), small molecule Mnk1/2 protein degraders that potently inhibit Mnk-eIF4E and mTORC1/4E-BP1/p70SK6 oncogenic signaling in breast and prostate cancer models [29,33,38,39,40]. NRs exert potent anticancer effects in breast and prostate cancer cells and tumor xenografts by inducing ubiquitin-proteasomal degradation of Mnk1/2 and preventing eIF4E and mTORC1 activation, thereby leading to inhibition of cancer cell growth, apoptosis evasion, cell migration and invasion, in vitro and inhibition of tumor xenografts, in vivo [29,33,38,39,40]. Among various NRs, lead racemic VNLG-152 (4-(±)-(1*H*-imidazol-1-yl)-*N*-(4-fluorophenyl)-(*E*)-retinamide; termed VNLG-152R; Figure 1A) exhibited exemplary Mnk-eIF4E/m-TORC1 inhibitory and in vitro anticancer activities in a variety of breast cancer subtypes [39].

Here, we present exciting data, for the first time, on VNLG-152R binding affinity to Mnk1 protein, impact on TNBC pro-inflammatory cytokines secretion, inhibition of Mnk-eIF4E and mTORC1/4E-BP1/p70SK6 signaling, in vivo toxicity profile, anti-tumor efficacy in MDA-MB-231 cell line derived xenograft (CDX) and TNBC patient derived xenografts (PDX) models, and anti-metastatic effects in vitro and in vivo. We also extended our studies to examine the functional activity of the two enantiomers of VNLG-152R (termed VNLG-152E1 and VNLG-152E2), compared with racemic VNLG-152R with respect to growth inhibition and Mnk-eIF4E signaling in TNBC cell subtypes, in vivo toxicity, pharmacokinetic in mice, and anti-tumor efficacy in TNBC xenograft models. Altogether, the results presented, especially the potent inhibition of MDA-MB-231 CDX and PDX TNBC tumor growth and metastasis in vivo, including robust in vivo targets engagement, and remarkable induction of apoptosis, with no apparent host toxicity, provides a strong scientific rationale for the development of racemic VNLG-152R as a novel therapeutic agent for TNBC, and possibly, other cancers and diseases driven by Mnk-eIF4E and mTORC1 signaling.

## 2. Results

### 2.1. VNLG-152R Interacts with Mnk1, Inhibits eIF4E Complex Formation and − Is Important for Its Antiproliferative Activity in TNBC Cells in Vitro

We have previously established that racemic VNLG-152R (Figure 1A) induces Mnk1/2 degradation (with concomitant depletion of peIF4E) to inhibit the growth of BC/TNBC cells by reducing proliferation and inducing apoptosis. We also showed that it was specific in inducing the degradation on Mnk1/2, no inhibition of Mnk1/2 kinase activities, and with no effect on the other components of the eIF4F complex, that is, eIF4G, eIF4E and eIF4A [39]. Furthermore, we showed that VNLG-152R did not have any significant effects on Mnk1/2 effectors (ERK/p38MAPK and protein phosphatase 2A, PP2A) or Akt/pAkt (potential mediators of Mnk1/2-eIF4E pathway) [39]. These data strongly suggest that VNLG-152R might be a specific Mnk1/2 degrader that inhibits cancer cell growth [33,39,40]. It should be noted that although VNLG-152R strongly degrades both Mnk1 and Mnk2, Mnk1 knockdown alone has been shown to be sufficient to decrease tumor formation in nude mice [11,41].

To further establish that Mnks are the prime targets of VNLG-152R, we focused on Mnk1, where molecular docking studies predicted the binding energy (∆G_binding_) of VNLG-152R with the ATPase domain of Mnk1 to be −6.1 kcal/mol. As shown in Figure 1B, VNLG-152R formed hydrogen bonds with Leu55 and Phe192, including three hydrophobic and one π-cation interactions with other amino acids. To obtain evidence supporting direct binding of VNLG-152R to Mnk1, we synthesized VNLG-152R-biotin conjugate (Figure 1C) to capture recombinant Mnk1 protein and Mnk1 protein in TNBC cells. We treated recombinant Mnk1 protein with VNLG-152-biotin or biotin and used streptavidin beads to pull-down biotinylated conjugates. As shown in Figure 1D, left panel, substantial amounts of Mnk1 was found only in the VNLG-152R-biotin-treated sample. In addition, we showed direct binding of Mnk1 to VNLG-152R-biotin in MDA-MB-231 cells (Figure 1D, right panel), suggesting that VNLG-152R-induced Mnk1 degradation may be by direct binding to Mnk1. Using surface plasmon resonance (SPR) assay (OpenSPR, Nicoya Lifesciences, Waterloo, ON, Canada) our scouting analysis showed that VNLG-152R (1.25, 10 and 30 µM) bound to Mnk1 in a concentration-dependent manner (Figure 1E), with a dissociation constant (*K_D_*) calculated to be 6.46 × 10^−6^ M. 

Taken together, these results strongly suggest that VNLG-152R interacts with Mnk1 protein. In future studies, we will investigate the binding/interaction of VNLG-152R with Mnk2. Mnk1 and Mnk2 are the only known physiological kinases of the eIF4E oncoprotein [42]. Having established that VNLG-152R specifically interacts and inhibits Mnk1/2 activities (via proteasomal degradation of Mnk1/2 proteins), we next evaluated the effect of VNLG-152R on eIF4E interaction with the mRNA cap structure in TNBC cells using the cap analog m^7^GTP (7-methylguanosine 5′-triphosphate) immobilized to Sepharose in a pull-down assay. As shown in Figure 1F, VNLG-152R (20 µM, 24 h) treatment resulted in decreased eIF4G and increased eukaryotic translation initiation factor 4E-binding protein 1 (4E-BP1) association with eIF4E. An increase in 4E-BP1 binding to eIF4E and a corresponding decrease in eIF4G association is indicative of translational repression (inhibition eIF4F complex formation).

To confirm that Mnk1 inhibition is a major target underlying the observed anticancer effects in TNBC cells, we investigated the impact of Mnk1 knockdown on cell survival in MDA-MB-231 cells by MTT assay. Figure 1G, left panel shows western blot analysis that confirmed the efficiency of siMnk1 transfection and Mnk1 knockdown, similar to our previously published data [33,39,40]. MTT assay revealed that transient transfection of MDA-MB-231 cells with Mnk1 siRNA for 24 h caused ~20% decrease in cell viability compared to control. Treatment of MDA-MB-231 cells with VNLG-152R (5 μM) showed a robust reduction (~75%) in cell viability. However, VNLG-152R (5 μM) treatment of MDA-MB-231 cells harboring Mnk1 knockdown did not show any significant decrease in cell viability (Figure 1G, right panel), These results strongly suggest that Mnk1 is indispensable for VNLG-152R to mediate its growth inhibitory effects in MDA-MB-231 cells, and that loss of a prime target abolishes the growth-inhibitory effects mediated by VNLG-152R.

### 2.2. VNLG-152R Inhibits mTORC1 Signaling and the Production of Pro-Inflammatory Cytokines

The novel and highly significant recent serendipitous finding that Mnk1/2 regulates mTORC1 signaling and associates with mTORC1 directly [31,32] suggest that our Mnk1/2 degraders can also inhibit mTORC1/4E-BP1/p70SK6 signaling. Indeed, our previous report [33] and other independent studies [34,35,36,37] strongly support the connection between Mnk1/2 and mTORC1. As we expected, VNLG-152R treatment of two TNBC cell lines (MDA-MB-231 and MBA-MB-468 for 72 h) caused strong dose-dependent depletion of p-mTOR (at Ser2448 site), p4E-BP1, peIF4E and p-p70 S6K (Figure 2A). The total proteins of these molecular targets were not affected by VNLG-152R (Figure 2A). Mechanistically, because phosphorylation of mTORC1 at the Ser2448 site is a reliable biomarker of its activity [43], this data is consistent with Mnk1/2 regulation of mTORC1 signaling [31,32]. It should be noted that the depletion of p-mTOR is not due to inhibition of its prime kinase, Akt, because we have previously reported that VNLG-152R does not inhibit Akt’s kinase activity, that is, no impact on the levels of both total Akt and pAkt^Ser473^ [39].

Two recent reports on Mnk1/2 inhibitors (BAY 1143269 inhibits only Mnk1, while eFT508 inhibits both Mnk1 and Mnk2) that are currently under clinical evaluation highlight the significance of Mnk inhibition of pro-tumorigenic cytokines secretions [23,44]. Thus, to enhance the translational potential of VNLG-152R, we conducted head-to-head studies comparing VNLG-152R versus eFT508. Using the well-established 3-(4,5-dimethylthiazolyl-2)-2,5-diphenyl tetrazolium bromide (MTT) assay, we show that VNLG-152R has remarkably more potent antiproliferative activities against MDA-MB-231 (GI_50_ values for VNLG-152R and eFT508 = 0.89 and 7.07 µM, respectively; 8-fold difference) and MDA-MB-468 (GI_50_ values for VNLG-152R and eFT508 = 0.50 and 52.46 µM, respectively; 105-fold difference) TNBC cells than eFT508 (Figure 2B).

We next expanded our study to unravel the impact of these two agents on growth factor and cytokine signaling pathways. Treatment of MDA-MB-231 TNBC cells with VNLG-152 or eFT508 led to dose-dependent reduction in secreted IL-6 and IL-8 (Figure 2C), TNFα (Figure 2D) and chemokine CXCL1 (Figure 2E), but clearly show that VNLG-152R is markedly more effective than eFT508, that seem to stimulate secretion of CXCL1. Similar results were also obtained using MDA-MB-468 cells. This promising data also suggests that VNLG-152R has a greater potential to reshape the TNBC tumor microenvironment (TME) in vivo.

### 2.3. VNLG152R and Its Enantiomers Exhibit Differential Antiproliferative Activities and Pharmacokinetics but Exhibit Equivalent Safety/Toxicity Profiles

Due to concerns with development of racemic drugs [45], it was important for us to characterize the pharmacological and toxicological properties of racemic VNLG-152R compared with its two enantiomers, VNLG-152E1 and VNLG-152E2. Indeed, the US Food and Drug Administration (FDA) [46] and the drug development industries [47,48,49,50,51] require rigorous characterization of racemates and their pure enantiomers of potential drugs to clearly establish differential activities/toxicities if any, or lack thereof. Enantiopure VNLG-152E1 and VNLG-152E2 were obtained by chiral HPLC separation (see Appendix A); the chromatograms for the pure enantiomers and that for the racemate from which they were obtained are presented in Figure 3A. Using the MTT assay, we clearly demonstrated (Figure 3B) that the enantiomers exhibited differential inhibition of proliferation of a panel of 6 human TNBC cell lines [MDA-MB-468 (BL1), HCC-1086 (BL2), MDA-MB-231/MDA-MB-436 (MSL) and MDA-MB-453/MFM-223 (LAR), representing 4 of the 6 subtypes] [52,53,54]. On average, VNLG-152E2 exhibit GI_50_ of 0.39 µM (range: 0.10–1.17 µM), whereas the average GI_50_ values of VNLG-152E1 and racemic VNLG-152R were 2.3 µM (range: 0.98–4.79 µM) and 0.97 µM (range: 0.62–1.86 µM), respectively (Figure 3B).

The plasma pharmacokinetics (PK) profiles were obtained after single intravenous (IV) and oral (PO) dosing of the three compounds in mice. Compounds concentrations in isolated plasma was quantified by liquid chromatography/tandem-mass spectrometry (LC-MS/MS). As shown in Figure 3C, top panel, the elimination kinetics for the three compounds were almost identical. Contrary to results from IV experiments, systemic exposure after oral dosing, as indicated by area under the curve (AUC) and bioavailability (%F), were not identical (Figure 3C, bottom panel and Figure 3D). The oral elimination half-lives (T_1/2_s) were good to excellent (4.02–9.25 h) and the three compounds showed high oral bioavailability (73–162%) (Figure 3D), supporting oral administration for further in vivo toxicity and antitumor efficacy studies.

Finally, to assess the in vivo toxicity profiles of VNLG-152R, VNLG-152E1 and VNLG-152E2, we attempted to determine the maximum tolerated dose (MTD) of the three compounds by administering groups of 3 female athymic nude mice (the strain used for the xenograft model) with three different oral doses of each compound (10, 25, 50 mg/kg/day) 5 consecutive days/week for 14 days. This was followed by observation for an additional 14 days. There were no abnormalities in the body weight kinetics, which were comparable in the control and compounds-treated animals throughout the study. At the doses used, none of the compounds cause weight loss, reduced food intake, behavioral changes, any other overt clinical signs of toxicity, or mortality (Figure 3E). Because we did not observe any toxicity/mortality at these doses, we estimate that the MTDs of these compounds are each >50 mg/kg. Furthermore, this data also suggests that multiple dosing of VNLG-152R and its enantiomer at 50 mg/kg/day is safe.

### 2.4. VNLG-152R Exhibited Potent Antitumor Activity in Cell-Line Derived and Patient Derived TNBC Xenograft Models and the Effect on Tumor Growth is Associated with Its Effects of Mnk1/2 and peIF4E, Downstream Targets and Induction of Apoptosis 

During the early stage of this study, we did not have ready access to required quantities of the enantiomers of racemic VNLG-152. Because we were curious to unravel the in vivo antitumor efficacy of our lead Mnk1/2 degrader, our initial studies were conducted using only VNLG-152R. We first tested the antitumor activity of VNLG-152R in the difficult-to-treat MDA-MB-231 cell-line derived tumor xenograft (CDX) model of TNBC. Thus, athymic nude mice bearing MDA-MB-231 tumor xenografts were treated (IP) with different doses of VNLG-152R [5, 10, or 20 mg/kg, dissolved in 40% β-cyclodextrin (BCD) in ddH_2_O] or vehicle (BCD), 5 consecutive days a week for 4 weeks. VNLG-152R at 10 and 20 mg/kg/day, were equipotent and most effective, resulting in a tumor growth inhibition (TGI) of 97% (*p* < 0.0001) in the mean final tumor volume compared with control (Figure 4A). It is also notable that VNLG-152R at a low dose of 5 mg/kg/day caused TGI of 91% (*p* > 0.0001 vs. control) (Figure 4A). It is important to state here that the control was part of a large study with our other retinamides whose data has already been reported [38]. Representative photos of the tumor-bearing mice in the VNLG-152R-treated versus control groups at termination of the experiment (day 28) are depicted in Figure 4B. Importantly, no significant changes in body weights were observed (Figure 4C), and no gross organ (liver, lung and kidney) abnormalities (Figure 4D), suggesting that the compound is safe at the effective doses.

To further validate the molecular mechanism(s) underlying the effects of VNLG-152R, we analyzed the association between the antitumor effects observed in vivo and the suppression of Mnk1/2 downstream signaling, the levels of peIF4E and downstream targets, cyclin D1 and apoptosis modulators (Bcl, Bax, and Bad). Indeed, western blot analysis of tumors treated with VNLG-152R at the two doses (10 or 20 mg/kg) evaluated showed significant dose-dependent suppressions of Mnk1, Mnk2, p-eIF4E, cyclin D1, Bcl-2, and up-regulation of Bax and Bad. As expected, no significant changes in the expression of total eIF4E was observed (Figure 4E). It is notable that the profound increase in the Bax/Bcl-2 ratio (quantified by densitometry) from ~0.09 (control) to 42.3 (VNLG-152R, 20 mg/kg group), i.e., 470-fold, is exceptional (Figure 4F). Immunostaining of the tumor tissues confirmed downregulation of Mnk1, Mnk2 and peIF4E expressions in VNLG-152-treated groups (Figure 4G).

We next tested the antitumor efficacy of VNLG-152R in a patient-derived primary TNBC (Br-001) xenografts (PDX) in mice. Br-001 was derived from an African American woman aged 74 years at University of Maryland Medical Center and propagated via serial transplantation in NSG mice Athymic nude mice bearing PDX, Br-001 tumor xenografts were treated (IP) with VNLG-152 10 mg/kg/day, 5 days a week, like the procedure described above for the MDA-MB-231 xenografts. VNLG-152R treatment led to significant tumor growth inhibition (TGI) of 73.3% (Figure 4H). No significant changes in body weights were observed (Figure 4I) and no organ toxicity was found, suggesting that the compound is safe at the effective dose. Gratifyingly, western blotting results also showed significant remarkable suppressions of Mnk1 and peIF4E and up-regulation of pro-apoptotic protein, cPARP (Figure 4J).

Collectively, our data show that VNLG-152R effectively suppressed the growth of two different TNBC xenograft tumors (CDX and PDX), with no apparent host toxicity. In addition, we show remarkable robust target/biomarker engagement, in vivo.

### 2.5. Racemic VNLG-152 Is Better than the Enantiomers at Inhibiting Tumor Growth and Mnk/eIF4E Signaling in TNBC Xenografts In Vivo

We subsequently validated the tumor growth inhibitory effect of racemic VNLG-152 and its enantiomers in nude mice bearing MDA-MB-231 xenografts, comparing head-to-head oral (PO) versus intraperitoneal (IP) injections at doses of 20 mg/kg per day for 36 days. As shown in Figure 5A,C, all 3 compounds exhibited significant tumor growth inhibitions with no clear difference between IP versus PO administrations. Except for VNLG-152E1 PO treatment with a TGI of 78%, the other treatments were considered almost equipotent (%TGI of 91–100%) because of the large variations in tumor volume that resulted to insignificant statistical differences between the treatment groups. However, it should be noted that treatments with the racemate caused both tumor growth inhibition and regression (see Figure 5C). No host toxicity was also observed in any of the treatment groups as monitored by changes in body weight throughout the study (Figure 5B). Treatment with the racemate and its enantiomers via both routes of administration significantly induced modulation of the expression of Mnk1, Mnk2, p-eIF4E, Cyclin D1, Bcl-2 and Bad compared to vehicle administered group. The modulatory effects on these proteins was more pronounced in groups administered VNLG-152R, and as expected, all treatments did not have any significant effect on the expression of total eIF4E (Figure 5D).

We next evaluated the effect of oral administration of 20 mg/kg VNLG-152R and VNLG-152E1 in NSG mice bearing PDX, TNBC tumor xenografts. We could not test VNLG-152E2 because of limited available amounts. Our results revealed that both racemate and E1 enantiomer led to significant TGIs of 82 and 79%, respectively (Figure 5E,G). No changes in body weights were observed (Figure 5F) and no organ toxicity was found, suggesting that the compound is safe at the effective dose. Here again, western blot analysis showed significant suppressions of Mnk1 and peIF4E, and up-regulation of pro-apoptotic protein, c-PARP in treated groups, although the racemate-treated group appear to be more effective (Figure 5H).

### 2.6. Racemic VNLG-152 Inhibits Cell Migration In Vitro and In Vivo

Finally, to further enhance the translation potential of our lead Mn1/2 degrader, we investigated the effects of VNLG-152R on TNBC cell migration and invasion using impedance-based real time cell invasion assay. We had previously reported that VNLG-152R is a potent inhibitor of TNBC and prostate cancer cell growth, migration and invasion, in vitro [33,39,40]. In this study, we observed that MDA-MB-231 cells in the 10% FBS treated group (control) began to invade earlier, compared with those treated with different doses of VNLG-152R (2.5, 5 and 10 μM). The invasive ability of the control group also remained higher until the end of the experiment. In contrast, the invasive abilities of the cells in VNLG-152R treated groups were significantly lower (Figure 6A). It is important to state that we had previously reported [39] that VNLG-152R did not decrease TNBC cell proliferation within the time frame of the cell migration assay, indicating that VNLG-152R suppresses migration independent of its effect on cell proliferation.

Epithelial–mesenchymal transition (EMT) is defined by the loss of epithelial characteristics and the acquisition of a mesenchymal phenotype. EMT produces cancer cells that are invasive, migratory, and exhibit stem cell characteristics that have metastatic potential [56]. To gain insight into the role of VNLG-152R in cell migration, we investigated whether the observed decrease in MDA-MB-231 cell migration/invasion upon treatment with VNLG-152R is a result of inhibition of EMT-related molecular markers. The results demonstrated that treatment of the MDA-MB-231 and MDA-MB-468 cells with VNLG-152R at different concentrations for 24 h led to concentration-dependent decreases in the expression of N-Cadherin, β-catenin, MMP-2, MMP-9, Snail, Slug, Vimentin and TCF/ZEB-1 (Figure 6B), consistent with our previous results in PC-3 and CWR22Rv1 prostate cancer cells [33,40].

To further validate the anti-metastatic effects of VNLG-152R in vivo, we performed experimental lung metastasis assay. MDA-MB-231-Luc breast cancer cells were intravenously injected into a tail vein of SCID mice. The animals were randomly divided into a treatment group and a control group after cell injection. VNLG-152R was administered by IP injection at doses of 20 mg/kg for 2 weeks. As determined using the in vivo imaging system (IVIS), the MDA-MB-231-Luc illumination signals in the lung tissues of control mice were significantly stronger than those in the VNLG-152R treated mice on days 7 and 14 (Figure 6C), indicating that the VNLG-152R treatment potently (~80% inhibition) decreased the metastasis of TNBC cells.

## 3. Discussion

Several in vitro and in vivo studies have clearly shown that the Mnk-eIF4E axis promotes BC transformation and tumor progression [12,13,14,26,28], findings that are also supported by clinical data [7,15,22,28,57,58]. Currently, the major strategy of inhibiting Mnk-eIF4E signaling is preventing eIF4E phosphorylation via inhibition of the enzymatic activities of Mnk1 and Mnk2 (reviewed in [30]). Expanding upon our previous studies with our lead Mnk1/2 protein degrader VNLG-152R, in a variety of breast cancer subtypes, in vitro [39], we focused on studying the impact of VNLG-152R and its two pure enantiomers, specifically in several in vitro human triple negative breast cancer (TNBC) subtypes and in vivo xenograft and metastasis models of TNBC. We also conducted head-to-head anti-TNBC efficacy evaluation of VNLG-152R and eFT508, an Mnk1/2 inhibitor in ongoing clinical trials [23], using in vitro models of TNBC.

In the present study, using various assays such as molecular docking, SPR assay, biotin–avidin and m7 GTP pull-down assays, we establish that VNLG-152R (Figure 1A) specifically bound to the Mnk1 protein (Figure 1B–E) and disrupts eIF4F complex formation (Figure 1F). Additionally, binding of VNLG-152R to Mnk1 might promote its ubiquitination (via recruitment of E-3 ligase(s)) and proteasomal degradation [33,39,40]. Further studies are needed to determine the exact E-3 ligase(s) implicated in VNLG-152R-induced Mnk1 protein degradation. In target validation studies, we also showed for the first time that short interfering RNA (siRNA) mediated knockdown of Mnk1 protein in MDA-MB-231 cells prevents further antiproliferative activity of VNLG-152R (Figure 1G), supporting the essential role of Mnk1 in mediating VNLG-152R’s anti-TNBC activity. Thus, our data support the scientific premise that loss of MDA-MB-231 cells sensitivity to VNLG-152 is most likely due to the suppression of its major molecular target, Mnk1. The modest cell growth inhibition caused by Mnk1 knockdown could be due to possible proliferative effect of Mnk2.

Unlike the recently reported Mnk1/2 inhibitor under clinical development, eFT508 [23], VNLG-152R in addition to inhibiting the Mnk-eIF4E axis, also caused potent inhibition of mTORC1 activity in MDA-MB-231 and MDA-MB-468 TNBC cell lines (Figure 2A). Our compound is by far superior to eFT508 with respect to anti-proliferative activity, in vitro (Figure 2B). The inhibition of mTORC1 activity is consistent will the recent finding that Mnk1/2 regulate mTORC1 signaling [33,34,35,36,37] and associate with mTORC1 directly [31,32]. It should be noted that Mnks accomplish such regulation independent of their canonical role in phosphorylating eIF4E [31,32,59]. In addition, we showed for the first time that VNLG-152R potently inhibited the secretions of multiple pro-tumorigenic cytokines and chemokines (IL-6, IL-8, TNF-α and CXCL1) (Figure 2C,D) and was remarkably more effective than eFT508. These results are of particular interest based on a comprehensive review article that highlight the important role of Mnk1/2 regulated cytokine production in cancer progression, metastasis and chemoresistance [60] and on the recent finding that inhibition of both IL6 and IL8 is necessary to inhibit TNBC growth in vivo [61]. Indeed, this mechanism of action (i.e., the potential to reshape the tumor microenvironment, TME) is likely contributing to the antitumor efficacy in both the xenograft tumors and metastatic models and will be the focus of future studies. Given the emerging findings which clearly show that agents that cause target protein degradation (target degraders) are more efficacious than agents that inhibit the target’s enzyme activity (target inhibitors) [62,63,64], we hypothesize that Mnk1/2 degraders would be superior to Mnk1/2 inhibitors.

In vitro biological evaluation of racemic VNLG-152R and its two enantiomers against six TNBC subtype cell lines indicate differential antiproliferative activities (Figure 3B). In contrast, in vivo studies in mice indicate different pharmacokinetics but equivalent safety/toxicity profiles (Figure 3C–E). This safety/toxicity data in addition to the finding that no interconversion of the enantiomers was observed in mice, suggest that any of these compounds can be developed with confidence [45,46,49,50,51]. These data provided the impetus for head-to-head evaluations of these agents in anti-tumor efficacy studies in TNBC animal models.

Previously, we demonstrated that VNLG-152R inhibits TNBC cell growth, colonization, migration, and invasion by perturbing eIF4E activation via enhancing Mnk1/2 degradation [39]. In the present study, we observed that VNLG-152R dramatically caused remarkable dose-dependent suppression of tumor growth in the difficult-to-treat MDA-MB-231 human breast cancer xenografts (Figure 4A,B) with no apparent host toxicity (Figure 4C,D). VNLG-152R also efficiently reduced tumor growth/volume in the patient derived TNBC xenograft model, which replicates the histopathology, tumor behavior, and metastatic properties of the original tumor (Figure 4H). In both MDA-MB-231 and PDX tumor xenograft models, reduction in tumor growth/volume upon administration VNLG-152R administration was associated with downregulation of the expressions of Mnk1/2 and peIF4E, and modulation of several other downstream proteins that are translated via eIF4E including cyclin D1, and apoptosis biomarkers, Bcl2, Bax, Bad and c-PARP (Figure 4E–G,J). These findings are in accordance with our earlier in vitro data and substantiate the fact that loss of Mnk1/2 forfeits the formation of phosphorylated eIF4E, which in turn affecting formation of active eIF4F translational complex and ensuing oncogenic translation inhibition [42,65].

Mnk1 and Mnk2 are typically co-expressed and can both serve to activate eIF4E by phosphorylating eIF4E at Ser209 [30,65,66]. The finding that VNLG-152R is an equipotent dual degrader of both Mnks (Figure 4E,G) is gratifying because it ensures that its use would avoid compensatory signaling by either isoform. Our results are also concordant with recent findings by other groups who established that inhibition of eIF4E phosphorylation by Mnk1/2 inhibitors suppressed growth of diffuse large cell B-cell lymphoma and TNBC MDA-MB-231 tumor xenografts [23], cell-line and patient-derived non-small cell lung cancer xenografts [44] and metastasis of KIT-mutant melanoma [67]. Furthermore, our finding that VNLG-152R potently suppress the expressions of peIF4E and p4E-BP1 in TNBC in vitro and in vivo is exciting as the ‘homogenous expressions’ of these oncogenes are proving to be remarkable cancer targets deemed to be more important with respect to therapeutic targeting than the current well-established oncogenic targets such as PI3k, pAkt, pmTOR, pMAPK, HER2, BRCA1/2, AR/AR-Vs and ERα that are ‘heterogeneously expressed’ in most tumors [68,69,70].

We also showed that VNLG-152E2 exhibited greater potency in inhibiting TNBC proliferation and expression of Mnk1 and peIF4E in TNBC cell subtypes compared to VNLG-152R or VNLG-152E1. However, in MDA-MB-231 and PDX tumors in vivo, racemate appears to be more effective than the two enantiomers in inhibiting the tumor growth, volume and Mnk-eIF4E signaling associated proteins (Figure 5A,C–E,G,H). Although the reason for the different in vivo efficacies is not exactly known at this time, it is plausible that the stereoselective differences in pharmacokinetic and/or pharmacodynamic parameters could have a major role towards this effect [45], hypotheses that will be investigated in future studies.

Mnk1/2 overexpression may be responsible for the drug resistance of human cancer cells, and may be associated with tumor invasion, tumor recurrence and metastasis. Recent study by Beggs et al., [71] showed that genetic knockout of Mnk1/2 impaired the migration of embryonic fibroblasts both in two-dimensional wound-healing experiments and in three-dimensional migration assays. The study also established that a selective inhibitor of Mnk1/2 strongly blocks migration of fibroblasts and cancer cells besides reducing the expression of vimentin, a marker of mesenchymal cells. In addition, Sonenberg and colleagues provided a rationale for targeting eIF4E phosphorylation in both cancer cells and cells that comprise the TME to halt metastasis and demonstrated the efficacy of this strategy using merestinib, and Mnk1/2 inhibitor [72]. In this study, we provide compelling evidence from both cell lines and animal studies that, in addition to its effects on the primary tumor, VNLG-152R also has anti-metastatic effects. We demonstrated that VNLG-152R repressed the invasion of MDA-MB-231 cells (Figure 6A). In addition, VNLG-152R treatment altered the expression levels of EMT-related proteins such as β-catenin, N-cadherin, MMPs, snail, slug, and vimentin (Figure 6B) that are involved in crucial metastatic pathways [73]. The results of our in vivo experimental metastasis assays confirmed that VNLG-152R suppressed tumor metastasis (Figure 6C), likely through the inhibition of cell migration. We propose that VNLG-152R can inhibit tumor cell metastasis chiefly/partly via promoting Mnk1/2 degradation thereby hampering oncogenic eIF4E signaling.

Two recent studies demonstrated that pharmacologic inhibition of eIF4E with ribavirin [26] or inhibition of eIF4E phosphorylation with a multikinase inhibitor that is a potent Mnk1/2 inhibitor, merestinib [72] potently prevented metastasis of highly aggressive mammary cancer 66c14 cells but exhibited modest/no effect on primary tumor growth. The fact that our studies demonstrate potent inhibition of both primary mammary tumor growth and metastasis, would suggest that Mnk1/2 degraders are more likely to achieve more favorable clinical responses than Mnk1/2 inhibitors in breast cancer patients. The potential advantage of induced target protein degradation versus inhibition include prolonged pharmacodynamic effects and ability to abrogate non-enzyme-dependent, i.e., scaffolding function of the protein that thwarts feedback activation(s) [63]. Furthermore, it is also possible that the ability of our Mnk1/2 degraders to effectively and simultaneously inhibit both Mnk-eIF4E and mTORC1 pathways (Figure 7) may also contribute to their remarkable antitumor and anti-metastasis efficacies in vitro and in vivo. Indeed, recent review articles highlight the significance of dual abrogation of Mnk1/2 and mTORC1 as a novel therapeutic approach for the treatment of aggressive cancers [74,75].

Our previous [33,39,40] and current studies show that by targeting Mnk1/2 degradation, we induce differential effects on the translation and expression of specific genes. Our Mnk1/2 degraders target the expression of specific oncogenic proteins such as Bcl2, cyclin D1, cyclin D3, cyclin E, CDC25C, N-cadherin, MMP-2/9, IL-6, IL-8, TNFα, CXCL1, snail and slug on which cancer cells depend to maintain their transformed phenotype, whereas the expression of housekeeping proteins such as β-actin and GAPDH was not affected. This approach has a clear advantage over the generic genotoxic conventional chemotherapy. Our Mnk1/2 degradation-mediated inhibition of translation is not reduced to one single oncogene as other approaches do, but multiple oncogenic proteins, which has the potential to circumvent the problem of resistance to treatments with targeted therapies [76,77].

## 4. Materials and Methods 

### 4.1. Reagents.

eFT508.HCl (6′-[(6-aminopyrimidin-4-yl)amino]-8′-methyl-2′Hspiro[cyclohexane-1,3′-imidazo-[1,5-a]pyridine]-1′,5′-dione hydrochloride (1:1); catalog #: A-6030) was purchased from Active Biochem Kowloon, Hong Kong, ChinaeFT508.HCl was in DMSO, and 10 nM aliquots were stored at −80 °C. The following commercially available antibodies were purchased from Cell Signaling Technology (Danvers, MA, USA): p-eIF4E (Cat# 9741), Mnk1 (Cat# 2195), eIF4E (Cat# 2067), Cyclin D1 (Cat# 2978), Bcl-2 (Cat# 4223), Bax (Cat# 5023), Bad (Cat# 9292), N-Cadherin (Cat# 13116), β-catenin (Cat# 8480), MMP-2 (Cat# 40994), MMP-9 (Cat# 13667), Snail (Cat# 3879), Slug (Cat# 9585), Vimentin (Cat# 5741), TCF/ZEB-1 (Cat# 3396S), β-actin (Cat# 3700) and GAPDH (Cat# 5174). Mnk2 (Cat# M0696) antibody was purchased from Sigma (St. Louis, MO, USA) and mouse and rabbit horseradish peroxidase (HRP)-conjugated secondary antibodies (Cat# 7076 and 7074) was purchased from Cell Signaling Technology.

### 4.2. Cell Culture Treatment and Western Blotting

MDA-MB-231, MDA-MB-468, MDA-MB-436, MDA-MB-453, HCC-1806, and MFM-223 cells were purchased from American Type Culture Collection (ATCC, Manassas, VA, USA), and authenticated at the Biopolymer/Genomics Core Facility, University of Maryland Baltimore. Cells were tested for mycoplasma contamination at the University of Maryland Marlene and Stewart Greenebaum Cancer Center (UMGCC) Translational Shared Service. Cells were cultured in the recommended media supplemented with 10% FBS and plated 1 day prior to treatment. Cells were treated with the indicated dose and time in the figure at 37 °C in 5% CO_2_. Protein lysates were harvested in RIPA lysis buffer (Sigma). Western blotting was done as described previously [39] with antibodies described above.

### 4.3. Cell Growth Assays

Cell growth inhibition assay was performed as described previously [39]. For cell growth experiment, cells were treated with the VNLG-152R, VNLG-152E1 or VNLG-15E2 for 7 days. MTT assay was performed at the end of the experiment. Results represent the mean ± standard deviation of three independent experiments.

### 4.4. In Vitro m7 GTP Pull-Down Assay

eIF4F complex in cell extracts was detected using affinity chromatography 7-methyl GTP (m7GTP)-Sepharose as described previously [33,56]. Briefly, MDA-MB-231 treated with 20 μM VNLG-152 for 24 h were lysed in buffer containing 20 mM Tris-HCl (pH 7.4), 150 mM NaCl, 1 mM EDTA, 1 mM EGTA, 1 mM β-glycerophosphate, 1 mM Na_3_VO_4_, 1% Triton X-100, 2.5 mM sodium pyrophosphate, and protease inhibitor cocktail, on ice for 15 min. After centrifugation at 12,000× *g* for 15 min at 4 °C, the supernatants were collected and incubated with m7GTP-Sepharose (GE Healthcare Bio-Sciences Corp./Amersham, Piscataway, NJ, USA) at 4 °C for 2 h with constant shaking. Beads were washed three times with lysis buffer and three times with 1× Phosphate Buffered Salin, (PBS). The samples were then denatured, and the supernatants were then loaded to SDS-PAGE for immunoblotting.

### 4.5. siRNA-Mediated Knockdown of Mnk1 Gene Expression

Small interfering RNA targeting the Mnk1 (siMnk1) gene and corresponding scramble siRNA sequence (siControl) were purchased from Ambion ((Waltham, MA, USA). Briefly, 3 × 10^5^ MDA-MB-231 cells were seeded into 6 cm cell culture dish (Corning®) and incubated for 24 h in for normal culture medium. Subsequently, cells were subjected to transfection with 100 nM of siMnk1 or siControl using the RNAi Human/Mouse Starter Kit _(Qiagen,_ (Qiagen, Hilden, Germany) following the manufacturer’s instructions for 48 h. For cell growth assay experiments, transfection complex was removed after 48 h, cells were washed twice with phosphate-buffered saline and replaced with normal growth medium. Twenty-four hour later, 5 μM of VNLG-152R was added to the growth media and harvested after 24 h. Non-transfected cells were also treated with 5 μM of VNLG-152R for 24 h. Protein silencing was confirmed by immunoblot analysis [40].

### 4.6. Real Time Cell Invasion Assay

Real-time measurement of cell invasion was performed using the xCELLigence RTCA DP device (ACEA Biosciences, San Diego, CA, USA) [78]. MDA-MB-231 cells were treated with VNLG-152 (2.5, 5, and 10 μM) for 24 h, cells being treated with VNLG-152 when they were plated. Subsequently, cells were seeded (75,000/well) onto 16-well microelectronic censored, two chamber Matrigel-coated trans-well plates (CIM-plates) containing the respective drug in serum-free condition. Medium containing 10% serum (chemo-attractant) was added to the bottom wells. Invasion of cells was measured from the interaction of cells with the electrodes on the bottom surface of top chamber and represented as a change in cell index (CI), an arbitrary unit derived from the relative change in electrical impedance across microelectronic sensor arrays. The electrical impedance was captured every 3 min for an experimental duration of 24 h. The rate of invasion is expressed as the CI or the change in electrical impedance at each timepoint. Values are expressed as the ±SEM of the duplicate wells from three independent experiments.

### 4.7. Streptavidin–Agarose Pull-Down Assay

The streptavidin–agarose pull-down assay was performed as described previously [79]. Biotin-VNLG-152R (50 µg) and biotin (50 µg; negative control) were preincubated with streptavidin agarose beads (Invitrogen, Waltham, MA, USA) overnight at 4 °C. The biotin-VNLG-152R-conjugated beads were washed three times with cold PBS and then incubated with recombinant full length Mnk1 proteins or cell lysates overnight at 4 °C. The bead-bound proteins were incubated and separated on a 10% SDS-PAGE and detected by western blot analysis for Mnk1.

### 4.8. Multiplex Supernatant Cytokine Analysis (Luminex)

Supernatants from the MDA-MB-231 (TNBC) cells in Fetal Bovine Serum (FBS) containing media were treated with various concentrations of VNLG-12R for 72 h, were collected, and various cytokines and growth factor concentrations (CLCX-1, IL-6, IL8 and TNF-α) were determined by Luminex multiplex assay kit (Millipore, Billerica, MA, USA) according to the manufacturer’s instructions. The expressions of secreted cytokines were read on Luminex 100 reader. The data was then analyzed using the BioPlex Software (BioRad, Hercules, CA, USA). The Luminex multiplex assays were conducted at the University of Maryland School of Medicine Cytokine Core Laboratory (Baltimore, MD, USA). 

### 4.9. Surface Plasmon Resonance (SPR) Analysis

Purified recombinant Mnk1 protein with GST tag (2.5 µg/mL) was fixed on GST sensor chip (Nicoya) by capture-coupling, then VNLG-152R (1.25, 10 and 30 µM) were injected (flow rate of 20 µL/min) sequentially into the chamber in running buffer (PBS-T with 10% DMSO). The interaction of Mnk1 with VNLG-152R was detected by Open SPRTM (Nicoya Lifesciences, Kitchener, ON, Canada) at 30 °C. The chip was regenerated with regeneration solution (10 mM glycine-HCL) at a flow rate of 150 µL/min. The close curve fitting to the sensograms were calculated by global fitting curves (1:1 Langmuir binding model). The data was retrieved and analyzed with TraceDraw software (Kitchener, ON, Canada).

### 4.10. Animal Study Approval

All animal studies were performed according to protocol #1215008 reviewed and approved by the Institutional Animal Care and Use Committee (IACUC) at the University of Maryland School of Medicine, Baltimore, MD, USA, (IACUC No. 125008, 15 January 2016) 

#### 4.10.1. In Vivo Anti-Tumor Studies in MDA-MB-231 TNBC Xenograft Model

Female athymic nude (Jackson Laboratory, Bar Harbor, Maine, USA) 5–6 weeks of age were housed under complete aseptic conditions, fed autoclaved pellets and sterile water ad libitum. Following a week of acclimatization, approximately 1 × 10^6^ MDA-MB-231 cells were inoculated into both flanks for MDA-MB-231 derived xenograft tumors. Tumor-bearing mice (tumor volume around 90–110 mm^3^) were randomized into various treatment groups (5 mice in each group; compounds formulated in vehicle) and treated as follows: (i) vehicle control (40% β-cyclodextrin in ddH_2_O, IP), (ii) VNLG-152R (5 mg/kg, IP, once daily), (iii) VNLG-152 (10 mg/kg, IP, once daily) and (iv) VNLG-152 (20 mg/kg, IP, once daily) 5 days/week for 29 days. Tumors were measured twice weekly with calipers and tumor volume was calculated by the formula: length × width/2 × 0.5 (mm^3^). Animals were also weighed weekly and monitored for general health status and signs of possible toxicity due to treatments. Mice were sacrificed after the indicated periods of treatment and tumors and organs excised. Tumors were divided and either flash frozen in liquid nitrogen or placed in 10% buffered formalin for western blot analysis, immunohistochemistry (IHC), and hematoxylin and eosin (H & E) staining. It is important to state here that the control was part of a large study with our other retinamides whose data has already been reported [38].

The second experiment designed to compare heal-to-head IP versus oral dosing (PO) and the efficacy of racemic VNLG-152 versus its two enantiomers was like that described above but consisted of eight groups. When the tumor volumes had reached about 100 mm^3^, the mice were randomly divided into 8 groups of 5 mice each. The two control groups received vehicle (IP or OP) and the other six groups received VNLG-152R (20 mg/kg, IP or PO, once daily), VNLG-152E1 (20 mg/kg, IP or PO, once daily), and VNLG-152E2 (20 mg/kg, IP or PO, once daily) five days/week. These treatments continued for 36 days and the tumors were measured and processed as described above.

#### 4.10.2. In Vivo Anti-Tumor Studies in TNBC Patient Derived Xenograft (PDX) Tumor Model

Female NSG (NOD.Cg-PrkdcscidIl2rgtm1Wjl/SzJ) mice (Jackson Laboratory) 5–6 weeks of age were housed under complete aseptic conditions, fed autoclaved pellets and sterile water ad libitum. Primary triple negative breast cancer tissue from a patient in University of Maryland Medical Center (74-year-old African American women) was excised during surgery, de-identified and then implanted into the mammary fat pad of female NSG mice. After expansion and viable freeze back, breast PDX (Br-001) was grown in three mice as a subcutaneous tumor. When tumors reached approximately 600–800 mm^3^, mice were euthanized, tumors excised aseptically and cut into 3 mm × 3 mm pieces and re-implanted subcutaneously in female NSG mice. The established xenografts were subsequently passaged from mouse to mouse to expand xenograft numbers. Tumor-bearing mice (tumor volume around 90–110 mm^3^) were randomized into various treatment groups (five mice in each group; compounds formulated in vehicle) and treated as follows: (i) vehicle control (40% β-cyclodextrin in ddH_2_O, IP once daily) and (ii) VNLG-152R (10 mg/kg, IP, once daily) 5 days/week for 15 days. Tumors were measured and processed as described for the MDA-MB-231 xenograft model.

The second PDX tumor model experiment was conducted to assess the efficacy of VNLG-152R versus VNLG-152E1 following PO administration. When the tumor volumes had reached about 100 mm^3^, the mice were randomly divided into three groups of five mice each. The control group received vehicle (PO) and the other two groups received VNLG-152R (20 mg/kg, PO, once daily), and VNLG-152E1 (20 mg/kg, PO, once daily) five days/week. These treatments continued for 15 days and the tumors were measured and processed as described above.

#### 4.10.3. Toxicology Study

Female Athymic nude mice were used for a 14 days chronic dose toxicity study in which the mice were given different doses of VNLG-152R and the enantiomers VNLG-152E1 and VNLG-152E2. Three different doses, i.e., 10, 25, and 50 mg/kg/day (formulated in 40% β-cyclodextrin in saline) of each compound were administered daily by oral gavage for 14 days. Each group consisted of 3 mice. Their body weight kinetics (loss in body weight, a clinical sign of toxicity) was observed during the 14 days of dosing following established procedures [55,80]. Changes in body weight were scored on a scale of 0–4 as follows: (1) Weight loss: (10 g = 4, 7–9 g = 3, 4–6 g = 2, 1–3 g = 1; severe = 3, moderate = 2, slight = 1, none = 0) [55].

#### 4.10.4. In Vivo Experimental Lung Metastasis Assay

The MDA-MB-231-luc cells were the kind gift of Dr. Stuart Martin (UMB). Cells were grown in DMEM with 10% FBS, hygromycin and G418 to subconfluence (~50–70%) and then trypsinized, washed, and resuspended in PBS. 1 × 10^6^ cells were injected IV in a volume of 100 µL. Mice were treated for 14 days with vehicle or VNLG-152 by IP injection at doses of 20 mg/kg. The luciferase signals in the mice were detected and photographed using an IVIS in vivo image system on days 0, 1 and 14 [81].

#### 4.10.5. Immunohistochemical Analysis.

Tumor samples were fixed in 10% buffered formalin for 12 h and processed conventionally. The paraffin-embedded tumor sections (4 μm thick) were heat immobilized and deparaffinized using xylene and rehydrated in a graded series of ethanol with a final wash in distilled water. Antigen retrieval was done in 10 mM citrate buffer (pH 6.0) in microwave followed by quenching of endogenous peroxidase activity with 3.0% H_2_O_2_ in methanol (vol/vol). Sections were then incubated with specific primary antibodies (1:500 dilution) and epitopes detected using the Ultra-sensitive ABC staining kit (Thermo Fisher Scientific, Waltham, MA, USA). The images were captured with an EVOS® FL Auto Imaging System (Life Technologies, Carlsbad, CA, USA)

#### 4.10.6. Tumor Lysate Preparation and Western Blot Analysis

Western blotting of tumor tissue lysates was performed according to standard protocol. Briefly, 30–50 μg protein per lysate was denatured with 2× sample buffer and resolved on 12 or 16% Tris–glycine gels by sodium dodecyl sulfate–polyacrylamide gel electrophoresis. Separated proteins were transferred onto nitrocellulose membrane by western blotting, and membrane was blocked for 1 h in blocking buffer and then incubated with specific primary antibodies (1:1000 dilution), followed by peroxidase-conjugated appropriate secondary antibody (1:3000 dilution). Finally, proteins were visualized by enhanced chemiluminescence detection and exposure to X-ray film. To confirm equal protein loading, membranes were stripped and re-probed with mouse monoclonal anti-β-actin primary antibody (Sigma).

### 4.11. Statistics

All in vitro experiments were repeated at least three times and reported as means with standard error where applicable. Western blot on in vivo samples were repeated in at least two different tumor sections from different animals. Statistical analyses were performed using Graph Pad software (San Diego, CA, USA). Student T-test and Analysis of variance (ANOVA) (Figure 1G; Figure 4A,H, Figure 5A,E) were used to determine the significance of deviations or lack thereof. Differences between groups were considered statistically significant at *p* < 0.05.

## 5. Conclusions

Taken together, these results clearly show that our most efficacious Mnk1/2 degrader, VNLG-152R exerts remarkable anti-tumor and anti-metastatic effects in in vivo TNBC models via inhibiting Mnk-eIF4E and mTORC1 signaling without any significant toxicity. Our results also demonstrate that VNLG-152R exhibit greater in vivo anti-tumor efficacy than its two enantiomers. In summary, we describe here the first potent and highly selective Mnk1/2 degraders with strong in vivo anti-tumor and anti-metastatic activity in TNBC models and robust in vivo target engagement. MTD studies further demonstrate VNLG-152R and its enantiomers are safe at dose levels with profound therapeutic benefit. Thus, considering the crucial roles of Mnk-eIF4E and mTORC1 signaling in cancer growth and progression, and the challenges posed by the TNBC, we believe that VNLG-152R may be an ideal novel small molecule therapeutics for the treatment of patients with primary/metastatic TNBC and potentially other breast cancer subtypes, including other malignancies with dysregulated Mnk-eIF4E/mTORC1 signaling.

## Figures and Tables

**Figure 1 cancers-11-00299-f001:**
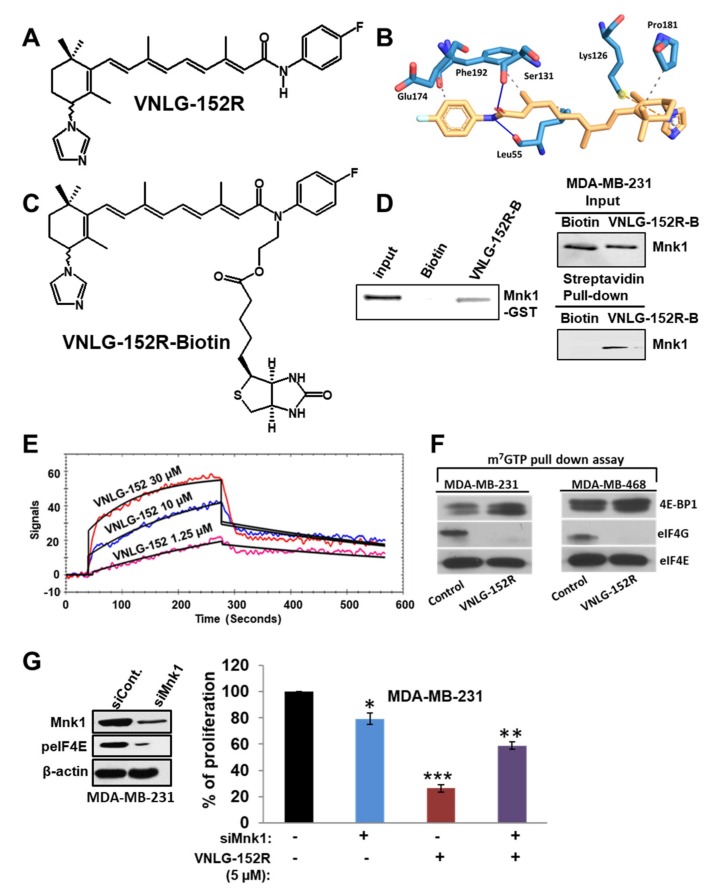
VNLG-152R interacts with mitogen-activated protein kinase (MAPK)-interacting serine/threonine-protein kinase 1 (Mnk1), inhibits eukaryotic initiation factor 4F (eIF4F) complex formation and is important for its antiproliferative activity in triple-negative breast cancer (TNBC) cells in vitro. (**A**) Chemical structure of VNLG-152R. (**B**) The Mnk1 ATP binding pocket and predicted interactions with VNLG-152R (yellow), yield a Gibb’s free energy of binding (∆Go of −6.1 kcal/mol. (**C**) The chemical structure of biotinylated VNLG-152R (VNLG-152R-biotin; VNLG-152R-B). (**D**) Purified recombinant GST-Mnk1 or MDA-MB-231 cells were treated with vehicle, biotin (50 µg) or VNLG-152R-B (50 µg) for 24 h. Recombinant protein or whole cell lysates were incubated with streptavidin beads overnight. Streptavidin pull-down and input control were subjected to immunoblotting with an antibody against Mnk1. (**E**) Competitive K_D_ determination of VNLG-152 to Mnk1 was determined using the surface plasmon resonance (SPR) assay (OpenSPR). Mnk1 was immobilized on a sensor chip. The assay was performed in triplicate. (**F**) Effect of VNLG-152R on eIF4E and eIF4G interaction in TNBC (MDA-MB-231 and MDA-MB-468) cells. Western blot analysis of eIF4G, eukaryotic translation initiation factor 4E-binding protein 1 (4E-BP1) and eIF4E from m7GTP purifications of vehicle and VNLG-152R (20 µM, 24 h)-treated cell lysates. (**G**) Effect of VNLG-152R (5 µM) on cell proliferation in MDA-MB-231 cells transfected with/without siMnk1 as determined by MTT assay (right panel). The results represent the mean ± SEM of three independent experiments and are represented as a bar graph after normalizing to control cells. *, *p* < 0.05; **, *p* < 0.01; ***, *p* < 0.001 compared with vehicle treated control. Western blot to confirm Mnk1 knockdown (left panel).

**Figure 2 cancers-11-00299-f002:**
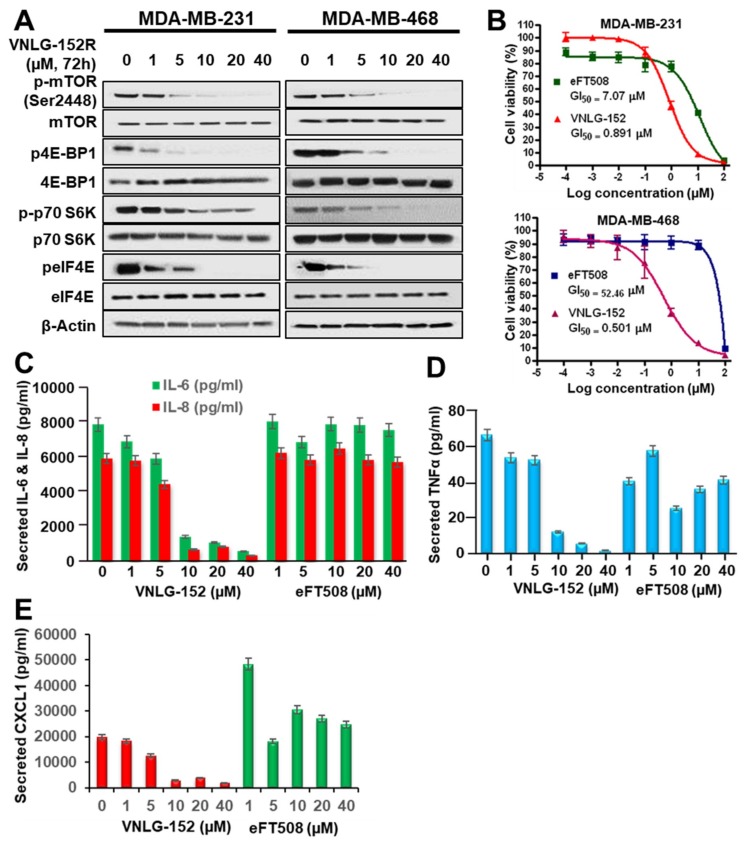
VNLG-152R inhibits mTORC1 signaling and the production of pro-inflammatory cytokines. (**A**) MDA-MB-231 and MDA-MB-468 cells were treated with the indicated concentrations of VNLG-152R for 72 h. Cell lysates were analyzed by immunoblotting with the indicated antibodies. (**B**) Head-to-head antiproliferative effects of VNLG-152R and eFT508 on MDA-MB-231 and MDA-MB-468 cells. The dose-response curves were generated from MTT assays after 6-day exposure to different concentrations of the compounds. Each point is a mean of replicates from three independent experiments. (**C**–**E**) MDA-MB-231 cell were treated with the indicated concentration of VNLG-152R or eFT508 for 72 h. Cell supernatants were collected, and the indicated cytokines were quantitated by Multiplex Luminex assay.

**Figure 3 cancers-11-00299-f003:**
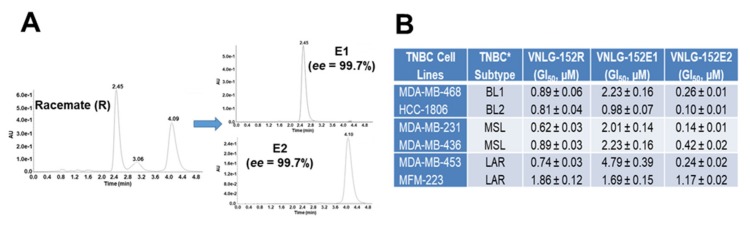
In vitro characterization, pharmacokinetics and toxicity profile of VNLG-152R and its two enantiomers. (**A**) HPLC chromatograms racemic VNLG-152R (left panel) and the resolved enantiomers VNLG-152E1 and VNLG-152E2 (right panel). The conditions of HPLC procedure are presented in Appendix A. (**B**) Table summarizing GI50 values (µM) each cell line treated with VNLG-152R and its enantiomers. TNBC cell subtypes were treated with listed compound (0.1 nmol/L–100 µmol/L) for 7 days and the GI50 values for the anti-proliferative effects of the compounds were determined from dose response curves (by a nonlinear regression analysis using Graph Pad Prism). Data represents the results from six independent experiments for each cell line. (**C**) Plasma concentration versus time profile after intravenous (IV, 2 mg/kg each, top panel) or oral (PO, 20 mg/kg each, bottom panel) of VNLG-152R (red), VNLG-152E1 (purple) and VNLG-152E2 (blue) to female CD1 mice. Values represent the mean plasma concentrations from three mice per time point. (**D**) Table of the pharmacokinetic parameters of various compounds after IV and PO dosing. AUC = area under the curve, T_1/2_ = elimination half-life, C_max_ = maximum plasma concentration, F (%) = oral bioavailability. (**E**) Effects of VNLG-152R, VNLG-152E1 and VNLG-152E2 on body weight and clinical observations at chronic dose termination of study. Three different doses, i.e., 10, 25, and 50 mg/kg/day (formulated in 40% β-cyclodextrin in saline) of each compound were administered daily by oral gavage for 14 days. Each group consisted of three mice. Loss in body weight, a clinical sign of toxicity was observed during the 14 days of dosing and the maximum tolerated dose (MTD) was estimated following established procedures. Changes in body weight were scored on a scale of 0–4 as follows as described under methods [55].

**Figure 4 cancers-11-00299-f004:**
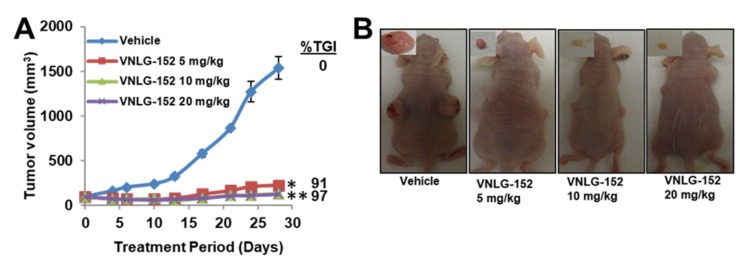
VNLG-152R potently suppress TNBC CDX and PDX tumor growth in vivo. (**A**) Tumor growth curves for subcutaneous MDA-MB-231 xenograft nude mouse model: nude mice bearing MDA-MB-231 xenograft tumors (on both flanks, five mice per group) were treated with VNLG-152R (5, 10 and 20 mg/kg, IP), 5 days per week for 29 days. (**B**) Representative mice with tumors from the vehicle-treated and VNLG-152R-treated groups on day 29. (**C**) Body weight kinetics of tumor bearing mice was assessed (once per week) throughout the study. (**D**) Representative images of hematoxylin-eosin staining of liver, lung and kidney organs from mice bearing MDA-MB-231 xenograft tumors (scale bar, 20 µm). (**E**) The effect of compounds on the expression of proteins modulated by Mnk/eIF4E signaling. Total cell lysates in mice treated with vehicle and different doses of VNLG-152R were prepared separately using RIPA buffer. Total protein (50 µg/well) from pooled samples (*n* = 5) was run on 10% SDS-PAGE and probed with antibodies for Mnk1, Mnk2, p-eIF4E, eIF4E, cyclin D1, Bcl-2, Bax, Bad and GAPDH. (**F**) Quantification Bax/Bcl-2 ratio of from E. (**G**) Representative images of Mnk1, Mnk2, and p-eIF4E immunostaining (left panel). The quantification of each molecular marker is shown on the right panel (scale bar: 20 µm). (**H**) Tumor growth curves for subcutaneous patient-derived TNBC xenograft nude mouse model: nude mice bearing Br-001 xenograft tumors (on one flank, five mice per group) were treated with VNLG-152R (10 mg/kg, IP), 5 days per week for 17 days. Insert show representative tumors from the vehicle- and VNLG-152R-treated groups after termination of the study on day 17. (**I**) Body weight kinetics of tumor bearing mice was assessed (once per week) throughout the study. (**J**) Effects of treatment on Mnk1, p-eIF4E, PARP and c-PARP were assessed by western blotting as described above for the MDA-MB-231 xenograft model. Results are represented as means ± SEM, *, *p* < 0.01; **, *p* < 0.001; compared with vehicle treated control.

**Figure 5 cancers-11-00299-f005:**
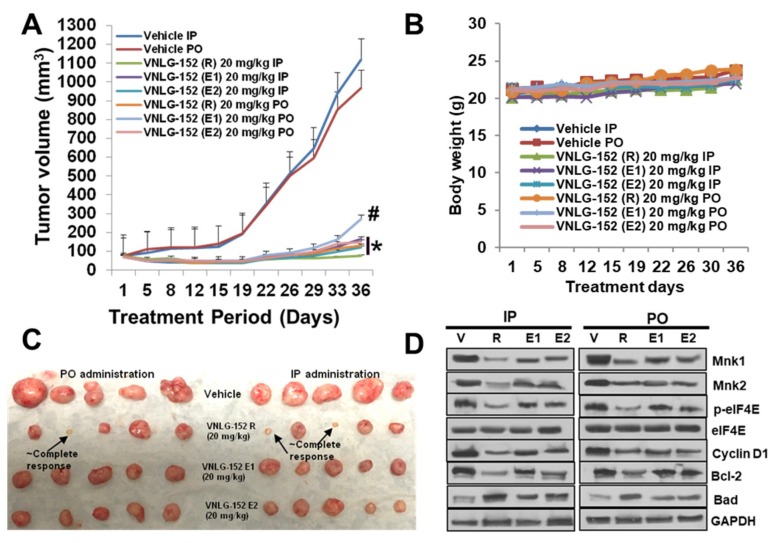
VNLG-152R and its enantiomers potently suppress TNBC CDX and PDX tumor growth in vivo, irrespective of the route of administration. (**A**) Tumor growth curves for subcutaneous MDA-MB-231 xenograft nude mouse model: nude mice bearing MDA-MB-231 xenograft tumors (on one flank, five mice per group) were treated with VNLG-152R, VNLG-152E1 or VNLG-152E2 (20 mg/kg, IP or PO), 5 days per week for 36 days. Tumor volumes were measured twice a week. (**B**) Body weight kinetics of tumor bearing mice was assessed (once per week) throughout the study. (**C**) Tumors from the vehicle-treated and compounds-treated groups on day 36. (**D**) Effects of treatment on Mnk1, Mnk2, p-eIF4E, eIF4E, cyclin D1, Bcl-2 and Bad were assessed by western blotting as described in Figure 4. (**E**) Tumor growth curves for subcutaneous patient-derived TNBC xenograft nude mouse model: nude mice bearing Br-001 xenograft tumors (on one flank, five mice per group) were treated with VNLG-152R or VNLG-152E1 (20 mg/kg, PO), 5 days per week for 15 days. (**F**) Body weight kinetics of tumor bearing mice was assessed (once per week) throughout the study. (**G**) Tumors from the vehicle-treated and compounds-treated groups on day 15. (**H**) Effects of treatment on Mnk1, p-eIF4E, eIF4E, PARP and c-PARP were assessed by western blotting as described above in Figure 4. Results are represented as means ± SEM, # *p* < 0.001; * & **, *p* < 0.0001, compared with vehicle treated control.

**Figure 6 cancers-11-00299-f006:**
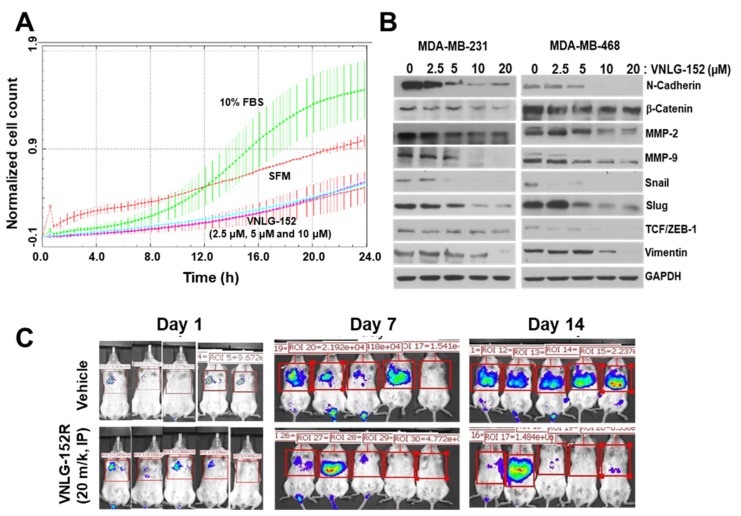
VNLG-152R inhibits TNBC cells metastasis in vitro and in vivo. (**A**) Real-time invasion assay with xCelligence device in MDA-MB-231 cells. MDA-MB-231 cells were pre-treated with VNLG-152R (2.5, 5, and 10 µM) for 24 h and 75 × 10^3^ cells were plated into 16 well matrigel coated CIM plate in the xcelligence device which measured invasion of cells across the filter every 3 min using electrical impedance across microelectric sensors. The rate of invasion is expressed as the cell index (CI) or the change in electrical impedance at each time point. Values are expressed as the ±SEM of the duplicate wells from three independent experiments. (**B**) The effects of VNLG-152R on the expression of EMT-related proteins in MDA-MB-231 and MDA-MB-468 cells. (**C**) About 1 × 10^6^ MDA-MB-231-Luc cells were intravenously injected into a tail vein of SCID mice. VNLG-152R (20 mg/kg, IP) was administered to mice, 5 days/week for 2 weeks. The luciferase signals in the mice were detected and photographed using an IVIS in vivo image system on days 1, 7 and 14.

**Figure 7 cancers-11-00299-f007:**
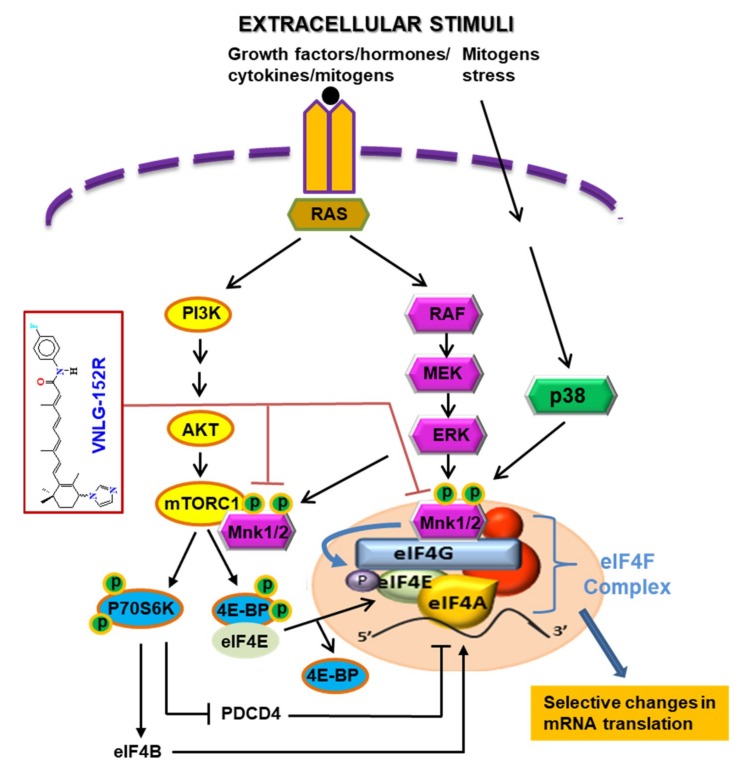
Schematic of the major signaling pathways (Mnk-eIF4E and mTORC1) affecting protein translation and how our Mnk1/2 degrader targets the two major pathways in TNBC. By targeting these two crucial oncogenic pathways, we posit that Mnk1/2 degraders will be efficacious against all types of tumor cells, regardless of their genetic composition.

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
