# Peer review of "The Novel Mnk1/2 Degrader and Apoptosis Inducer VNLG-152 Potently Inhibits TNBC Tumor Growth and Metastasis"

_cancers, 2019, doi:10.3390/cancers11030299_

Round 1

Reviewer 1 Report

General Comments:

This is generally a well-designed study and the manuscript is well-written.

Specific Comments:

1.     Figure 1, VNLG-152R interact with Mnk1, how about Mnk2? Dose the drug also bind with Mnk2? Any difference in the predictive binding model structurally? (Figure 1B)

2.     Figure 1G, the results of si-Mnk1 RNA treatment with or without VNLG-152R showed that si-MnK1 RNA alone exerts less anti-proliferative effect compared to VNLG-152R alone, and si-Mnk1 RNA somehow counteracts with VNLG-152R, suggesting indeed VNLG-152R needs Mnk1 to act as anti-cancer drug. However, it also raised the question that since VNLG-152R acts via Mnk1 degradation, why the effects of protein silencing exerts less anti-proliferative effect? The authors should provide explanation in Discussion.

3.     Figure 3C, the pharmacokinetic (PK) data suggests oral route should have a slower drug elimination (longer T1/2), however, Figure 3D table demonstrates the opposite, can the authors confirm?

4.     Figure 4D and Figure 4G contains imaging issue (4D, liver H.E stain pictures, 4G, Mnk1 IHC figures, please compare with prior publication [Fig. 1C and 2A] in FEBS J. 2018 Mar;285(6):1051-1063), which needs correction.

5.     Figure 6A legend: “75 x 103 cells” should be 75 x 103 cells?, 6C legend: About 1x106 MDA-MB-231 Luc cells should be 1 x 106 ?

Author Response

A.             Response to Comments:

Reviewer #1:

Q1: Figure 1, VNLG-152R interact with Mnk1, how about Mnk2? Does the drug also bind with Mnk2? Any difference in the predictive binding model structurally? (Figure 1B).

Response: In this study we focused on Mnk1. We will investigate the interactions/binding of VNLG-152R with Mnk2 in future studies. This statement is included in the revised manuscript (please see page 4, lines 147-148).

Q2: Figure 1G, the results of si-Mnk1 RNA treatment with or without VNLG-152R showed that si-Mnk1 RNA alone exerts less anti-proliferative effect compared to VNLG-152R alone, and si-MNK1 RNA counteracts with VNLG-152R, suggesting indeed VNLG-152R needs Mnk1 to act as an anti-cancer drug. However, it also raised the question that since VNLG-152 acts via MNK1 degradation, why the effects of protein silencing exert less anti-proliferative effect? The authors should provide explanation in discussion.

Response: Herein this experiment, only Mnk1 was transiently knocked down in MDA-MB-231 cells (using siMnk1). Both activated Mnk1 and Mnk2 contribute to cell proliferation. We anticipate that, even with loss of Mnk1 via gene knock down, activated Mnk2 could still promote cell proliferation in MDA-MB-231 cells. This might be the reason for the less anti-proliferative effect observed in MDA-MB-231 cells that were treated with siMnk1 alone. As suggested, this explanation has now been included in the results section of revised manuscript (please see page 16, lines 468-469).

Q3: Figure 3C, the pharmacokinetic (PK) data suggests oral route should have a slower drug elimination (longer T1/2), however Figure 3D table demonstrates the opposite, can the authors confirm.

Response: We apologize for this error and thank Reviewer #1. We have revised Fig 3D accordingly

Q4: Figure 4Dand 4G, contain imaging issues (4D, liver H&E stain pictures, 4G, Mnk1 IHC figures, please compare with prior publication (Fig. 1C and 2A) in FEBS J. 2018 Mar; 285(6):1051-1063), which needs correction.

Response: Here again, we apologize for this error and thank Reviewer #1. The first author conducted these liver H&E stain and Mnk1 IHC studies and those published in our FEBS J. 2018 Mar; 285(6):1051-1063) and in error used the incorrect figures. We have revised Fig 3D accordingly. Precise images for both H&E and IHC have now been included (Figure 4D & 4G).

Q5: Figure 6A legend: “75 x 103 should be 75 x 103? 6C legend: About 1x 106 MDA-MB-231-231 Luc cells should be 1 x 106 ?,

Response: These are a typographical error; these has been corrected in the revised manuscript) (please see page 15, lines 438 and 443).

We strongly believe that we have adequately responded to their various concerns and hope that our revised manuscript is now suitable for publication in Cancers. Thank you.

Dr. Vincent C. O. Njar,

Professor

A.             Response to Comments:

Reviewer #1:

Q1: Figure 1, VNLG-152R interact with Mnk1, how about Mnk2? Does the drug also bind with Mnk2? Any difference in the predictive binding model structurally? (Figure 1B).

Response: In this study we focused on Mnk1. We will investigate the interactions/binding of VNLG-152R with Mnk2 in future studies. This statement is included in the revised manuscript (please see page 4, lines 147-148).

Q2: Figure 1G, the results of si-Mnk1 RNA treatment with or without VNLG-152R showed that si-Mnk1 RNA alone exerts less anti-proliferative effect compared to VNLG-152R alone, and si-MNK1 RNA counteracts with VNLG-152R, suggesting indeed VNLG-152R needs Mnk1 to act as an anti-cancer drug. However, it also raised the question that since VNLG-152 acts via MNK1 degradation, why the effects of protein silencing exert less anti-proliferative effect? The authors should provide explanation in discussion.

Response: Herein this experiment, only Mnk1 was transiently knocked down in MDA-MB-231 cells (using siMnk1). Both activated Mnk1 and Mnk2 contribute to cell proliferation. We anticipate that, even with loss of Mnk1 via gene knock down, activated Mnk2 could still promote cell proliferation in MDA-MB-231 cells. This might be the reason for the less anti-proliferative effect observed in MDA-MB-231 cells that were treated with siMnk1 alone. As suggested, this explanation has now been included in the results section of revised manuscript (please see page 16, lines 468-469).

Q3: Figure 3C, the pharmacokinetic (PK) data suggests oral route should have a slower drug elimination (longer T1/2), however Figure 3D table demonstrates the opposite, can the authors confirm.

Response: We apologize for this error and thank Reviewer #1. We have revised Fig 3D accordingly

Q4: Figure 4Dand 4G, contain imaging issues (4D, liver H&E stain pictures, 4G, Mnk1 IHC figures, please compare with prior publication (Fig. 1C and 2A) in FEBS J. 2018 Mar; 285(6):1051-1063), which needs correction.

Response: Here again, we apologize for this error and thank Reviewer #1. The first author conducted these liver H&E stain and Mnk1 IHC studies and those published in our FEBS J. 2018 Mar; 285(6):1051-1063) and in error used the incorrect figures. We have revised Fig 3D accordingly. Precise images for both H&E and IHC have now been included (Figure 4D & 4G).

Q5: Figure 6A legend: “75 x 103 should be 75 x 103? 6C legend: About 1x 106 MDA-MB-231-231 Luc cells should be 1 x 106 ?,

Response: These are a typographical error; these has been corrected in the revised manuscript) (please see page 15, lines 438 and 443).

We strongly believe that we have adequately responded to their various concerns and hope that our revised manuscript is now suitable for publication in Cancers. Thank you.

Dr. Vincent C. O. Njar,

Professor

Reviewer 2 Report

Ramalingam et al. clearly show the anti-cancer effect of VNLG-152 both in vitro and in vivo. To improve the manuscript, some points should be addressed.

<Comments>

1. In Figure 2C-D, the authors analyzed the secretion levels of cytokines in the cells treated with VNLG-152 for 72 h. Please show the viability of cells treated with 10, 20 and 40 μM of VNLG-152.

2. The authors investigated the difference of VNLG-152E1 and E2. This is interesting. Please investigate their effects on Mnk1/2. 

Author Response

A.             Response to Comments:

Reviewer #2:

Q1: In Figure 2C-D, the authors analyzed the secretion levels of cytokines in the cells treated with VNLG-152 for 72h. Please show the viability of cells treated with 10, 20 and 40 µM of VNLG-152.

Response: We did not evaluate the cell viability of the MDA-MB-231 cells following treatment with VNLG-152R for 72h in this study. However, it should be noted that we have previously published data of MDA-MB-231 cell viability in cells treated with 0.1 nM – 100 µM of VNLG-152R for 6 days (144h) (ref. 39 of this manuscript). Based on our published data we are confident that the ≥ 50% of the MDA-MB-231 cells were viable following treatment with 10, 20 and 40 µM of VNLG-152.

Q2: The authors investigated the difference of VNLG-152E1 and E2. This is interesting. Please investigate their effects on MNK1/2.

Response: The effects of VNLG-152 E1 and E2 alongside VNLG-152R on the protein expression of Mnk1, Mnk2, p-eIF4E, total eIF4E and downstream targets of Mnk1/2 signaling activation such as cyclin D1, Bcl-2 and Bad have been investigated in TNBC xenografts in vivo (Figure 5D). As reported in this manuscript, because VNLG-152R show the most potent in vivo antitumor efficacy and because we did not observe and difference in the toxicities of VNLG-152R and its enantiomers, E1 and E2 in mice, we have elected to develop VNLG-152R.

We strongly believe that we have adequately responded to their various concerns and hope that our revised manuscript is now suitable for publication in Cancers. Thank you.

Dr. Vincent C. O. Njar,

Professor

Round 2

Reviewer 1 Report

The revised version has addressed prior reviewer's comments.

Reviewer 2 Report

No comment.